# When Viruses Meet Fungi: Tackling the Enemies in Hematology

**DOI:** 10.3390/jof8020184

**Published:** 2022-02-13

**Authors:** Alessandro Busca, Francesco Marchesi, Chiara Cattaneo, Enrico Maria Trecarichi, Mario Delia, Maria Ilaria Del Principe, Anna Candoni, Livio Pagano

**Affiliations:** 1Stem Cell Transplant Unit, AOU Citta’ della Salute e della Scienza, 10126 Torino, Italy; 2Hematology and Stem Cell Transplant Unit, IRCCS Regina Elena National Cancer Institute, 00144 Rome, Italy; francesco.marchesi@ifo.gov.it; 3Hematology, ASST-Spedali Civili, 25123 Brescia, Italy; chiara.cattaneo@asst-spedalicivili.it; 4Infectious and Tropical Diseases Unit, Department of Medical and Surgical Sciences, University “Magna Graecia”—“Mater Domini” Teaching Hospital, 88100 Catanzaro, Italy; em.trecarichi@unicz.it; 5Hematology and Stem Cell Transplantation Unit, AOUC Policlinico, 70124 Bari, Italy; mario.delia74@gmail.com; 6Department of Biomedicine and Prevention, Tor Vergata University of Rome, 00133 Rome, Italy; del.principe@med.uniroma2.it; 7Division of Hematology and Stem Cell Transplantation, University of Udine-ASUFC, 33100 Udine, Italy; anna.candoni@aoud.sanita.fvg.it; 8Unità di Ematologia Geriatrica ed Emopatie Rare, Fondazione Policlinico Universitario A. Gemelli—IRCCS, Università Cattolica del Sacro Cuore, 00168 Rome, Italy; livio.pagano@unicatt.it

**Keywords:** viral infections, invasive fungal infections, immunocompromised host, stem cell transplantation

## Abstract

The association of invasive fungal infections (IFI) and viral infections has been described in patients with hematologic malignancies (HM), in particular in hematopoietic stem cell transplant recipients. Regrettably, the diagnosis is often challenging, making the treatment inappropriate in some circumstances. The present review takes into consideration the viral infections commonly associated with IFI. Clinical presentation of IFI and viral infections, risk factors, and impact on the outcome of HM patients are discussed throughout the paper.

## 1. Introduction

Opportunistic infections represent a major hindrance to the successful outcome of patients receiving high-dose chemotherapy and stem cell transplantation for the treatment of hematologic malignancies (HMs). There is an extensive knowledge of invasive fungal infections (IFI) and viral infections as single complications in hematologic patients, while the relationship between these two entities is still far from being investigated in detail. The association of IFI and virus is not completely unexpected: the two infectious complications share common risk factors, such as high-dose chemotherapy, prolonged neutropenia, and lymphopenia, the presence of graft-versus-host disease (GVHD), and related treatments including high-dose steroids [1,2]. Several viral infections have been reported to be associated with an increased incidence of IFI; for instance, cytomegalovirus (CMV), community-acquired respiratory virus, and, more recently, severe acute respiratory syndrome coronavirus 2 (SARS-CoV-2) [3,4,5,6,7,8].

The present review aims to evaluate the currently available literature on the relationship between viral infections and IFI in patients affected by HMs.

## 2. Cytomegalovirus and Invasive Fungal Infections

CMV reactivation or, less frequently, primary infection is a typical complication occurring in immunocompromised patients, such as HMs. This complication has been extensively studied and is extremely relevant in patients undergoing allogeneic hematopoietic stem cell transplantation (allo-HSCT) [9]. On the contrary, outside allo-HSCT setting, the incidence of CMV reactivation was historically considered low, but the recent extensive use of pleiotropic immunosuppressive treatments and novel targeted agents in lymphoproliferative diseases led to a consideration of this viral complication as clinically relevant even in non-transplanted hematologic malignancy (HM) patients [10].

## 3. Incidence and Impact on the Outcome of Allo-HSCT Recipients

In allo-HSCT settings, CMV reactivation is often associated with end-organ disease (CMV disease) resulting in pneumonia in most cases and, more rarely, in a gastro-intestinal, hepatic, retinal, or central nervous system disease. At the present, the incidence of CMV disease is only 2–3% based on randomized clinical trials [11,12,13,14,15] and ranges between 5% and 10% in real-life practice [16,17]. Donor source (haploidentical and cord blood transplant), use of post-transplant high-dose cyclophosphamide, GVHD, and CMV serological status of patients and donors strongly influence the risk of post-transplant CMV reactivation and disease. CMV-related mortality significantly decreased over the last years, due to the introduction of pre-emptive treatment [9] and Letermovir prophylaxis [15].

## 4. Incidence and Impact on Outcome Outside Allo-HSCT Setting

CMV reactivation in HM patients treated with high-dose chemotherapy, immunosuppressive drugs, or novel targeted agents, is less frequent with a variable rate of CMV reactivation, ranging between 2% and 41% [18,19,20,21,22,23,24,25,26]. Overall, high-dose steroids, advanced disease status, poor performance status, and treatment with Rituximab, Alemtuzumab, Bortezomib, Fludarabine, and Idelalisib were the main risk factors for CMV reactivation in these clinical settings. In most cases, CMV reactivation in HM patients outside the allo-HSCT setting is easily manageable and CMV end-organ disease is a rare complication.

## 5. Association CMV-IFI

Several studies have so far been published about CMV and IFI in HMs, most of them focusing on patients undergoing allo-HSCT. Table 1 shows the most relevant published studies addressing this issue on adult HMs in the last 20 years. Many studies focusing on allo-HSCT setting suggested that CMV infection was an independent risk factor for IFI. However, the results showed in these studies were quite heterogeneous and not fully comparable, being obtained in retrospective study cohorts, having different methodological designs, and evaluating variously the CMV outcome. In particular, the association between IFI and CMV end-organ disease was evaluated in only four studies [27,28,29,30]. In all these studies a CMV end-organ disease was an independent risk factor for IFI: transplanted HM patients were exposed to a 3–7-fold higher risk of developing an early or late IFI. In the remaining nine studies, the authors focused on CMV DNAemia or antigenemia. In seven out of these nine studies [31,32,33,34,35,36], a significant association between CMV viremia and early or late IFI occurrence was found, whereas the remaining two [37,38] failed to demonstrate a significant association. Of note, the only prospective cohort study among all those analyzed, published by Atalla and coworkers, showed that CMV reactivation was a significant risk factor for the late occurrence of invasive mold infections (hazard ratio: 5.5), together with a diagnosis of lymphoma and prolonged neutropenia [32]. Outside allo-HSCT, there are fewer published studies. In a recent study about patients suffering from lymphoproliferative malignancies undergoing autologous hematopoietic stem cell transplantation (ASCT), Marchesi et al., showed a strong correlation between CMV symptomatic infection and early occurrence of IFI. However, a cause-effect relationship was not demonstrated in this study [3]. Moreover, Stanzani et al., reported a significant relationship at univariable but not at multivariable analysis between CMV and invasive mold infections in a large retrospective study on 1695 hospitalized HM patients [39]. The proper mechanisms leading to the increase of IFI in patients affected by CMV reactivation/end-organ disease are far from being well characterized and understood. Putative mechanisms may be neutropenia induced by anti-CMV treatments (i.e., ganciclovir), but also the immunosuppressive effect of CMV itself, sustained by the impairment of macrophages migration and antigen presentation [40]. As expected, the most frequently reported fungal infections were those caused by *Aspergillus* species, particularly invasive pulmonary aspergillosis (IPA). The reason for this lies of course in the higher incidence of IPA among HM patients compared to other fungal infections; however, some studies have suggested that the local viral inflammation in the pulmonary tract may lead to IPA occurrence [29,41].

## 6. Impact on the Outcome

The impact on clinical outcome of both CMV infection and IFI has been largely studied and well-characterized in HM patients when considered as single entities. However, the specific impact of concomitant CMV reactivation/end-organ disease and IFI remains to be determined. In allo-HSCT setting, Yong et al., showed a significant impact of IFI on overall survival irrespective of the presence of a GVHD; on the contrary, no survival differences occurred if patients did or did not experience CMV reactivation, with the only exception of those in which an acute GVHD was diagnosed [4]. However, in this study, no data about the clinical impact of concomitant CMV reactivation/disease and IFI were reported. In a retrospective study on allo-HSCT setting published in 2014, Gimenez et al. reported slightly significant higher mortality in patients with concomitant active CMV infection and invasive aspergillosis (IA) compared with patients affected only by IA [39]. Moreover, some other data showed a negative impact of pulmonary CMV co-infection in patients with *Pneumocystis jirovecii pneumonia* (PJP) [42]. To the best of our knowledge, there are still no specific published data on this issue in HMs outside the allo-HSCT setting. Further studies are warranted to better establish the real impact of CMV-IFI co-infection on morbidity and mortality of HM patients in both allo-HSCT and outside allo-HSCT setting.

## 7. Key Points

-Among hematologic patients, allogeneic HSCT recipients are at high risk for CMV-IFI co-infection-CMV infection emerged in several studies as an independent risk factor for the occurrence of IFI.-Neutropenia induced by CMV treatments, and the immunosuppression related to CMV infection itself may be considered as the main factors underpinning the IFI co-infection

## 8. Influenza and Invasive Fungal Infections

Influenza (A and B) viruses are single-stranded enveloped RNA viruses belonging to the Orthomyxoviridae family. The Influenza A virus subtype is the most common human pathogen with two major antigens on the viral capsid surface: the hemagglutinin (subtypes H1 to H18) and the neuraminidase (subtypes N1 to N11), of which minor or major genetic variations in the corresponding RNA genome occur each year. Influenza B virus is less prone to antigenic variation. The Influenza virus is transmitted from person to person via the respiratory route (with small-particle aerosols generated by infected humans) spreading across the population during the fall-to-winter and winter-to-spring months (the “so-called” flu season). Worldwide, 3–5 million people (2% to 10% of the global population) develop a severe influenza infection every year, leading to 50,000–100,000 deaths annually in Europe and USA and, of the hospitalized patients, 5–10% needs ICU admission [43,44]. The most severe and problematic influenza complication is the lower respiratory tract infection (LRTI), with localized or diffuse pneumonia. The incidence of influenza infection in patients with HM and allo-HSCT recipients has been reported between 1,3 and 40% according to the hematologic patient population and the laboratory diagnostic methods [45]. In HMs, progression from upper to LRTI occurs after a median of 1 week of infection onset, and recent studies suggest that approximately 30% of influenza infections in patients with leukemia and allo-HSCT recipients are complicated by concomitant influenza pneumonia. In addition, some studies have highlighted an association between the degree of lymphopenia and the development of influenza-related pneumonia, particularly in allo-HSCT patients [44].

Bacterial super-infection, mainly with *Streptococcus pneumoniae* and *Staphylococcus aureus*, is a well-known complication of severe influenza and influenza-related pneumonia. However, in recent years, an increasing number of publications on influenza-associated IFI are reported, mainly IPA (Influenza-Associated Pulmonary Aspergillosis-IAPA), particularly in patients with influenza A infection [43,44,46,47]. In these cases, the most frequently associated underlying conditions were immunosuppressive therapy for a variety of underlying diseases or procedures (including allo-HSCT), HM, and diabetes [43]. A recent study of the Dutch-Belgian Mycoses Study Group (DB-MSG) evaluated the incidence of IAPA in a large cohort of 432 patients (of whom 117 were immunocompromised) admitted to the ICU with severe influenza. In this study, a total of 83 of the 432 patients (19%) were diagnosed with IAPA, and the 90-day mortality was 51%, which was higher than the mortality in the 349 patients without IAPA (28%; *p* < 0.001) [48]. In addition, the multivariate analysis showed that the emergence of IAPA was independently associated with 90-day mortality, as were age, APACHE II score, diabetes, and immunocompromised status according to EORTC/MSG criteria [48]. Other similar epidemiological studies confirmed, in Europe and Asia, an incidence of IAPA ranging between 12% to 28% in patients with severe influenza and influenza-related pneumonia with a very high related mortality [48,49].

The diagnosis of IAPA is frequently performed early after ICU admission (5–7 days after admission) based on results of fungal culture from sputum or tracheal aspirates and/or on Galactomannan antigen (GM) detection in serum and, preferentially, in bronchoalveolar lavage (BAL) [50]. A diagnostic bronchoscopy is recommended to collect BAL, to look for tracheobronchitis, and to biopsy evident lesions. If IAPA is excluded on admission, but progressive radiological and/or clinical deterioration is observed during or after ICU admission, a repeated radiological and/or bronchoscopy evaluation is needed to exclude IAPA again.

To decrease the risk of progression to LRTI, an early recognition and treatment of influenza infection is crucial in HM, particularly in allo-HSCT recipients. In this patient population, antiviral treatment should be started as early as possible. Neuraminidase inhibitors (NAIs) are the first-line agents for the treatment of influenza A or influenza B. The currently available NAIs include oral oseltamivir, inhaled zanamivir, and very recently intravenous peramivir. The standard dose of oseltamivir for treatment of influenza is 75 mg twice daily for 5 days. In the cases with IAPA is indicated to treat these patients, according to guidelines for the treatment of IPA, with voriconazole (current gold standard therapy for IPA according to ECIL/ESCMID-ECMM-ERS/IDSA guidelines) [43,44] or liposomal amphotericin B. Other interventions, including nebulized antifungal therapy in cases with invasive *Aspergillus* tracheobronchitis, can also be considered to achieve the best therapeutic antifungal drug concentrations at the site of infection.

In clinical practice, influenza vaccination of HMs is strongly recommended by international guidelines as the primary method of influenza, and severe related complications, prevention (including IAPA). Patients with HM are reported to have low immunogenicity of influenza vaccine, particularly those who receive SCT and drugs such as rituximab, which depletes memory B cells [51]. Although the immune response to vaccination could be partial, vaccination remains the backbone of prevention of influenza infection and its complications (including IAPA) in HM patients [43,44].

## 9. Key Points

-Influenza infection is a cause of significant morbidity and mortality in patients with HM, particularly those undergoing chemotherapy and recipients of allo-HSCT.-Close monitoring for the development of IAPA and bacterial pneumonia is very important in managing hematologic patients with influenza-A prompt antiviral therapy is associated with reduced morbidity and mortality attributable to influenza in patients receiving chemotherapy or allo-HSCT-Influenza vaccination remains the most crucial prevention strategy, even if efficacy may be limited in immunocompromised hosts.

## 10. Respiratory Viruses and Invasive Fungal Infections

Respiratory virus infections (RVIs) include a large number of viruses—namely, Parainfluenza (PIV), Respiratory Syncytial Virus (RSV), Adenovirus (AdV), Human Metapneumovirus (HMPV), Human bocavirus (HBoV), Coronaviruses (CoVs) other than SARS-CoV-2, and Rhinovirus (HRV). RVIs are increasingly reported as a cause of significant morbidity and mortality in recipients of allo-HSCT and patients with HM [52,53]. Based on the widespread use of molecular diagnostics, the epidemiology and spectrum of clinical manifestations of these infections might be better characterized, thus deciphering the RVIs other than influenza in terms of incidence and clinical outcomes. Furthermore, the possible association between fungi and RVIs remains an expected data, although largely misunderstood.

Because of the broad range of diagnostic strategies (i.e., multiplex polymerase chain reaction (PCR), viral culture, direct antigen fluorescence), as well as the defined seasonality of RVIs, the incidence varies from 1 up to 10, 20, 30, 40, and 50% for HPMV, CoV, PIV+ Adenovirus and RSV, respectively. Of note, the mortality rate is by no means negligible [45]. In particular, the detection of RVIs in the immediate pre-transplant period is associated with LRTI and a decreased survival at 100 days, in particular in symptomatic patients [54,55]. Progression from URTI to LRTI is associated with an increased likelihood of fatal outcomes [54,55,56,57]. Consequently, attention should be paid to the factors correlating with the development of LRTI; for instance, older age, lymphopenia, early post-transplant infection, myeloablative conditioning regimen, HSCT from mismatched donors, and GVHD necessitating high-dose immunosuppressive agents [52,56,58]. In addition, risk factors for increased mortality are mismatched allogeneic HSCT, high-dose steroids at the time of LRTI, oxygen requirement, and mechanical ventilation [54,57,59,60,61,62].

## 11. Association RVIs and IFI

RVI seems to be an important risk factor for the development of IA in allo-HSCT, although it is difficult to define the contribution of viral infection among the large spectrum of conditions favoring immunosuppression [29]. It has been postulated that RVIs in the early posttransplant period make the lungs a target for allo-immunity thus contributing to the development of idiopathic pneumonia syndrome, bronchiolitis obliterans syndrome, and bronchiolitis obliterans organizing pneumonia [63]. As a result, coinfection with other viruses and superinfection with bacteria or fungi are described in allo-HSCT recipients with RVI [54,57,63,64,65,66,67]. In particular, Hardak et al. [66] reported that allo-HSCT and GVHD were predictors of polymicrobial infections: 8 pulmonary IA episodes out of 27 (30%) RVIs were documented with BAL. In a series of hematological and immunosuppressed patients who were investigated with BAL for suspected PJP, the coinfection was documented in 2 out of 5 (40%) Pneumocystis documented infections. [67] Ajmal et al., reported 4 cases of coinfections in immunocompromised patients, although 2 out of these cases were possible IFI [68]. Interestingly, in a series of 54 post-RVI IPA infections (including 14 cases of influenza), the coinfection was demonstrated to occur early, even within one week after the diagnosis of RVIs [46].

Clinical symptoms vary according to the spectrum of illness ranging from paucisymptomatic to respiratory failure, in URTI and LRTI, respectively. Regarding imaging investigations, it is often difficult to distinguish between RVIs and other infectious and/or noninfectious entities. Imaging may vary from diffuse bilateral ground-glass infiltrates, small multifocal nodules, bronchial vascular thickening, and/or airspace consolidation [69,70,71,72]. Regrettably, it should be emphasized that roughly half of the immunocompromised patients have a radiographic abnormality at the time of diagnosis but there are no features able to discriminate coinfections (RVIs + IPA) from IPA alone [46]. Only a retrospective study of CT findings in HCT recipients found that those infected with RSV were more likely to have an airway-centric pattern of disease with tree-in-bud opacities and bronchial wall thickening with or without consolidation. Adenovirus appeared as multifocal consolidation or ground-glass opacities without inflammatory airway findings [73], although these features are suggestive, rather than highly specific.

The outcome of patients with RVI and IFI remains particularly dismal, especially in allo-HSCT recipients where the coinfection is mostly reported (Table 2).

## 12. Key Points

-Scattering data on IFI associated with RVI have been reported in the literature.-Allo-HSCT recipients represent the subjects with the highest risk of coinfection with fungi and RVIs-The diagnosis may be demanding due to a lack of specific radiologic features.

## 13. SARS-CoV-2 and Invasive Fungal Infections

### General Population

IFI has been described as severe complications of the clinical course in patients with Coronavirus disease 19 (COVID-19), in particular in ICU settings. The wide use of broad-spectrum antibiotics, corticosteroids and/or immunomodulatory drugs, and central venous catheters have been hypothesized as the main risk factors for IFI in COVID-19 patients; besides these, comorbidities (e.g., chronic lung diseases, diabetes mellitus) and the direct pulmonary damage caused by SARS-CoV-2 have been reported as favoring factors [74,75,76,77].

Invasive yeast infections have been described in patients with COVID-19 infection, with candidiasis by far the most frequently reported infection, albeit with a high variability of incidence rates ranging from 0.7 to 23.5% [75,78]. Although a role of SARS-CoV-2 has been hypothesized in favoring invasive candidiasis during COVID-19, directly by a cellular injury or indirectly by immunological dysregulation, a clear association between these two infectious diseases has not been demonstrated to date [75,78].

Invasive mold infections have been described as associated or at high incidence in COVID-19 patients [78]. Singh et al., have recently reviewed 101 reported cases of mucormycosis in non-HM patients with active COVID-19 or post-COVID-19 course [79]. Most of these cases suffered from diabetes mellitus (80%) and had received corticosteroids (76.3%); the infection involved mainly nose and sinuses (88.9%), but also rhino-orbital localization was frequently reported (56.7%); interestingly, 81% of these cases were reported in India, thus indicating an important role of local epidemiology in the pathogenesis of the disease; the overall mortality was 30.7% [79].

Moreover, IPA is reported to have a high incidence in patients with severe COVID-19 and treated in ICU. Mitaka et al., recently conducted a systematic review and meta-analysis including 28 observational studies (conducted mainly in Europe) with a total of 3148 ICU patients diagnosed with COVID-19 to estimate incidence and mortality of COVID-19-associated pulmonary aspergillosis (CAPA); they reported an estimated incidence of CAPA in the ICU of 10.2% (range 2.3–34.4%); however, CAPA was associated with a mortality rate as high as 54.9% (range 22.2–94.4%) [77].

Several factors have been highlighted as predisposing to CAPA. These include the severe lung damage due to COVID-19 course, the use of corticosteroids (and/or immunomodulators) in patients with respiratory failure and mainly in those with acute respiratory distress syndrome (ARDS), the large use of broad-spectrum antibiotics, the presence of pulmonary chronic underlying diseases, and immunological alterations during COVID-19, such as leukopenia or lymphopenia and immune dysregulated response [75,76].

The mechanisms underlying the pathogenesis of CAPA have not been exactly understood, with only a few hypotheses. Firstly, corticosteroids could reduce the production of pro-inflammatory cytokines involved in an immune response against *Aspergillus* spp., mainly by suppressing the production of Nuclear Factor-kB (NF-kB), and on the other hand, increase the production of other cytokines (e.g., IL-10) which are responsible for inhibition of phagocytosis of the fungus [80]. Secondly, a possible role of molecules released by damaged pulmonary cells during COVID-19, i.e., danger-associated molecular patterns (DAMPs), which could display a signaling endogenous activity responsible for a dysregulated immune response leading to a higher risk for fungal infections in general, and CAPA in particular [75]. Thirdly, a direct role of SARS-CoV-2 by up-regulation of IL-1 pathway has been hypothesized on promoting CAPA [74]. Finally, to date, it is not clear if the immunomodulators (in particular, IL-6 antagonists) currently indicated in the treatment of patients suffering from COVID-19 and with ARDS could promote CAPA onset [74].

Diagnosis and management of CAPA present peculiar aspects concerning IPA in immunocompromised patients—in particular, those suffering from HMs; instead, clinical and prognostic characteristics of CAPA seem to be more similar to influenza-associated pulmonary aspergillosis (IAPA). IAPA and CAPA share the following characteristics: high prevalence, mainly in ICU settings, absence of classic risk factors for invasive mold infections (e.g., severe immunosuppression), presence of lymphopenia, and lack of classic radiological findings. In this regard, it should be underscored that there is large heterogeneity in diagnostic clinical (e.g., specific case definitions or diagnostic algorithms, such as EORTC/MSG definitions, AspICU algorithm) [8] and microbiological criteria (e.g., a cut-off of galactomannan index on serum or bronchoalveolar lavage) used among the studies reporting cases of CAPA and published to this date, as reported in the systematic review and meta-analysis of Mitaka et al. [77]. Importantly, this heterogeneity could have caused important biases in estimated prevalence and mortality rates reported. To provide specific criteria for CAPA diagnosis and recommendations for its management, a consensus statement was drafted by experts and endorsed by the European Confederation of Medical Mycology (ECMM) and the International Society for Human and Animal Mycology (ISHAM) [8]. Two different algorithms for the diagnosis of proven, probable, or possible pulmonary and tracheobronchial forms of CAPA have been purposed, including histological, microbiological, clinical, and radiological criteria. Voriconazole and isavuconazole have been recommended as first-line drugs for CAPA treatment; however, since multi-triazole resistance in *Aspergillus* spp. complicating cases of critical COVID-19 pneumonia have been reported, susceptibility tests for antifungal drugs (in particular for azoles), as well as therapeutic drug monitoring are recommended [81,82].

## 14. Hematological Malignancies

Patients with HM have always been considered at risk for fungal infections, due to the known immunosuppression both for the underlying diseases and for the specific treatments. It is known that fungal infections can worsen the prognosis of HMs. Indeed, adverse outcomes have been reported also in COVID-19 HMs, with related mortality higher than 30% [82,83,84,85,86,87,88]. However, epidemiologic data concerning COVID-19 HMs often lack information about possible co-infections. Only a few studies report data about co-infections or secondary infections in COVID-19 HMs, with findings often elusive. Among these, Sanchez–Pina et al. [89] reported a series of 39 patients, with no cases of bacterial or fungal co-infections, while Wu et al. [90] reported a not otherwise specified mixed co-infections in 8 out of 14 COVID-19 hematologic patients. In the Italian cohort study [91], no specified additional infections occurred in 187/536 (35%) COVID-19 hematologic cancer patients.

Therefore, specific information about fungal co-infections, mainly aspergillosis, in HM patients often refers to case reports or small series [91,92,93,94,95,96,97,98], which show the highly unfavorable prognosis of fungal co-infection also in most of this subset of patients. Only two cases of mucormysosis in HM COVID-19 patients have been reported, as the main risk factors for concurrent mucormycosis in COVID-19 patients seem to be diabetes mellitus and chronic renal failure [99]. Table 3 summarizes single cases of COVID-19 hematologic cancer patients with fungal co-infections.

Very recently, a multicenter cohort study of cancer patients with COVID-19 describing the epidemiology, risk factors, and clinical outcomes of co-infections and super-infections has been published [100]. In this large series, hematologic patients represent 43.8% of all cancer patients enrolled. Although co-infection at diagnosis and nosocomial super-infection were documented in 7.8% and 19.1% of patients, only one case of invasive pulmonary aspergillosis and two cases of polymicrobial bacteremia including *Candida* spp. were documented. Three cases of *Aspergillus* spp. tracheobronchitis were also documented. Reasons for this low incidence may be the retrospective nature of the study, the fact that diagnostic tests were performed according to the discretion of the physicians, and the lower sensitivity of biomarker tests in non-neutropenic patients. Regardless, fungal co-infections have been reported to significantly increase the risk of death in COVID-19 hematologic cancer patients [101].

As the COVID-19 pandemic could last a long time, hematologists should be aware of the risk of superinfections in COVID-19 hematologic cancer patients, particularly in those neutropenic and treated with steroids. A systematic workup of diagnostic tests for fungal infections should be always performed in these patients.

## 15. Key Points

-COVID-19-associated pulmonary aspergillosis (CAPA) has been well described in patients admitted in ICU, while very few data are available in hematologic patients.-The prognosis of hematologic patients with COVID-19 is extremely poor with mortality rates approaching 30% in many studies, and the outcome is even worse among patients with CAPA-Hematologic patients with COVID-19 infection should be investigated thoroughly for the possible coexistence of IFI.

## 16. Conclusions

Viral infections and IFI as single entities are well-described complications in patients affected by hematologic malignancies. By contrast, the coexistence of virus and fungi may represent a deceptive clinical issue. A great variety of viruses have been reported to be associated with fungi, almost exclusively represented by mold infections. Among HMs, HSCT recipients are the subjects at the highest risk of viral and IFI co-infections. Early interventions in terms of an accurate diagnostic work-up are of upmost importance to unveil this complication and adopt an appropriate therapeutic strategy. Whenever feasible, prioritization of vaccination programs is of paramount importance, although hematologic patients are not expected to generate robust responses. Further studies including a larger cohort of hematologic patients are eagerly needed to assess the impact of viral and IFI co-infections on the outcome of the patients.

## Figures and Tables

**Table 1 jof-08-00184-t001:** Studies on CMV-IFI association on adults HM patients in the last 20 years.

Reference	Type of Study	Patients	Setting	Year	Findings
Kimura et al. [31]	Retrospective	394	AlloHSCT	2020	In the multivariable analysis, grade II-IV acute GVHD and high CMV-AUC were identified as independent significant factors associated with a higher incidence of invasive mold infections
Marchesi et al. [20]	Retrospective	347	ASCT	2019	A strong correlation was found at univariable analysis between IFD and CMV symptomatic infection/end-organ disease
Yong et al. [4]	Retrospective	419	AlloHSCT	2017	CMV was and independent significant risk factor for late onset IFD
Shi et al. [30]	Retrospective	408	AlloHSCT	2015	A significant correlation between CMV disease and late onset IFD was found
Atalla et al. [32]	Prospective	345	AlloHSCT	2015	CMV reactivation was found to be a significant risk factor for late invasive mold infections, together with a diagnosis of lymphoma and neutropenia
Girmenia et al. [37]	Retrospective	1858	AlloHSCT	2014	CMV infection was not significantly associated with a risk of proven/probable IFD
Gimenez et al. [38]	Retrospective	167	AlloHSCT ^	2014	The occurrence of CMV DNAemia was not significantly associated with the subsequent development of invasive aspergillosis
Stanzani et al. [39]	Retrospective	1695	HM	2013	CMV reactivation was associated to the risk of invasive mold infection at univariable but not at multivariable analysis
Li et al. [33]	Retrospective	190	AlloHSCT	2012	Steroid treatment, CMV antigenemia and neutropenia were independent predictors of invasive mold infections
Zhang et al. [34]	Retrospective	286	AlloHSCT	2010	The multivariable analysis identified two risk factors for IFD: use of high-dose steroids and CMV infection
Mikulska et al. [35]	Retrospective	306	AlloHSCT	2009	CMV reactivation is associated with the risk of early onset invasive aspergillosis, together with active malignancy at transplant and delayed lymphocyte engraftment
Martino et al. [27]	Retrospective	219	AlloHSCT	2009	CMV disease was an independent risk factor for invasive aspergillosis together with steroid treatment and occurrence of lower respiratory tract infection
Garcia-Vidal et al. [28]	Retrospective	1248	AlloHSCT	2008	CMV disease was independently and significantly related to the risk of invasive mold infections
Martino et al. [29]	Retrospective	129	AlloHSCT ^^	2006	CMV was found to be a significant risk factor for invasive aspergillosis
Fukuda et al. [36]	Retrospective	163	AlloHSCT	2003	CMV found to be a risk factor for invasive mold infections at multivariable analysis, together with acute and chronic GVHD

Abbreviations. CMV: cytomegalovirus; IF: invasive fungal infection; AlloHSCT: allogeneic hematopoietic stem cell transplant; GVHD: graft versus host disease; ASCT: autologous hematopoietic stem cell transplant; HM: hematologic malignancies. ^ T-cell deplete alloHSCT; ^^ previous diagnosis of invasive aspergillosis.

**Table 2 jof-08-00184-t002:** Main studies reporting the association of respiratory virus infections and invasive fungal infections.

Reference	IFI/RVI (%)	No. alloHSCT/RVI (%)	Mortality Rate (%)	Comments
Parody et al. [56]	3/15 (20)	6/15 (40)	100	All IPA, retrospective study, reported coinfections in RVI
Martino et al. [29]	7/20 (35)	20/20 (100)	not reported *	All IA, prospective study on allotransplant patients
Ustun et al. [57]	19/173 (11)	134/173 (77)	67	13 IPA out of 19 fungal infections, retrospective study on transplant patients (auto and allo)
Ajmal S et al. [68]	4/4 (100)	2/4 (50)	25	1 case of IA, 1 probable Histoplasma, 2 possible IFI, retrospective study, reported only coinfections cases
Magira et al. [46]	54/54 (100)	33/54 (61)	45	All IPA, retrospective study, reported only IPA post RVI

Abbreviations. RVI indicates respiratory virus infection; IA, prover or probable invasive aspergillosis; IPA, invasive pulmonary aspergillosis; * mortality rate in IA: 48%.

**Table 3 jof-08-00184-t003:** Case reports of COVID-19 hematologic cancer patients with fungal coinfections.

Reference	Age/Sex	Hematologic Disease	Type of Fungal Coinfection	Level of Respiratory Assistance	Outcome
Blaize et al. [91]	74/M	MDS	Pulmonary aspergillosis	Mechanical ventilation	Died
Rutsaert et al. [7]	75/M	AML	Pulmonary aspergillosis	Mechanical ventilation	Died
Alanio et al. [92]	47/M	Myeloma	Pulmonary aspergillosis	Mechanical ventilation	Died
Spadea et al. [93]	19/M	ALL, allo-HSCT	Pulmonary aspergillosis	Non-invasive ventilation	Alive
Nasri et al. [94]	42/F	AML	Pulmonary aspergillosis	Non-invasive ventilation	Died
Falces-Romero et al. [95]	72/M 71/M	MDS CLL	Pulmonary aspergillosis Pulmonary aspergillosis	Not reported Not reported	Died Died
Höllein et al. [96]	69/F	sAML	Fungal pneumonia	Not reported	Died
Zurl et al. [97]	53/M	sAML	Pulmonary mucormycosis	Mechanical ventilation	Died
Bellanger et al. [98]	55/M	Lymphoma	Pulmonary apsergillosis and mucormycosis	Mechanical ventilation	Died

Abbreviations. MDS: myelodisplastic syndrome; AML: acute myeloid leukemia; ALL: acute lymphoblastic leukemia; CLL: chronic lymphocytic leukemia; sAML: secondary AML.

## Data Availability

Not applicable.

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
