# Peer review of "When Viruses Meet Fungi: Tackling the Enemies in Hematology"

_jof, 2022, doi:10.3390/jof8020184_

Round 1

Reviewer 1 Report

Comments to the Author

Comments on the manuscript: When viruses meet fungi: tackling the enemies in hematology by Busca et al.

The manuscript approaches a review about co-infections viruses and fungi in hematological populations.

Below are some corrections:

  1. Microorganism names must be written in italic. 
  2. The references cited in the text are with different numeration about topic "References".
  3. Revise the text punctuation.
  4. The citations of the references in the text must be according to the guide for authors of the journal.
  5. Microorganism genus must be written in capital. i.e the correct is Pneumocystis documented infections.
  6. Cite in the text only the author’s last name.
  7. Put legend in table 3.
  8. Candida spp/Candida spp.
  9. Standard the subtitles of the manuscript. The capital first letter of words or Capital first letter of the sentence.

Author Response

We thank the Reviewer for the comments to our paper. We tried to address all the issues raised by the Reviewer, and we do believe that the modifications contributed to improve the quality of the manuscript

Reviewer 1.

  1. Microorganism names must be written in italic. 

Names of microorganisms have been written in italic: all the modifications are marked in red.

  1. The references cited in the text are with different numeration about topic "References".

Numeration of references has been corrected

  1. Revise the text punctuation.

Text has been revised according to the suggestion of the Reviewer: all the modifications are marked in red.

  1. The citations of the references in the text must be according to the guide for authors of the journal.

References have been adjusted according to the Journal indications

  1. Microorganism genus must be written in capital. i.e the correct is Pneumocystis documented infections.

Text has been revised according to the indication of the Reviewer: all the modifications are marked in red.

  1. Cite in the text only the author’s last name.

The text has been modified according to the indication of the Reviewer

  1. Put legend in table 3.

Abbreviations have been added to table 3

  1. Candida spp/Candida spp.

The text has been modified according to the indication of the Reviewer

  1. Standard the subtitles of the manuscript. The capital first letter of words or Capital first letter of the sentence.

The text has been modified according to the indication of the Reviewer: capital first letter of words.

Reviewer 2 Report

Congratulations on this very interesting and well written review. I have only minor comments

-line 32: a word is missing between "completely unexpected" and "the two infectious complications" (as?)

-please check the format and numerotation of references, lots of errors/modification needed

--two references "1" Pagano 2017 and Ljungman 2011?

--doi is shown for only a few (2, 7, 44, 46, 48-49,52,53,78,80-81 and others) 

--some are underlined (47,49,88, 90, 91-93, 99, 100, 105, 107)

--journal in Italic (5) 

-- reference 6 : incomplete ++

--reference 7 = 85 , twice the same

--some references with 3 authors et al, others with more than 6 authors listed...

Author Response

We thank the Reviewer for the comments to our paper. We tried to address all the issues raised by the Reviewer, and we do believe that the modifications contributed to improve the quality of the manuscript

Congratulations on this very interesting and well written review. I have only minor comments

-line 32: a word is missing between "completely unexpected" and "the two infectious complications" (as?)

The text has been modified according to the indication of the Reviewer: “The association of IFI and virus is not completely unexpected: the two infectious complications share common risk factors, such as high-dose chemotherapy, prolonged neutropenia, and lymphopenia, the presence of graft-versus-host disease (GVHD), and related treatments including high-dose steroids”

-please check the format and numerotation of references, lots of errors/modification needed

numeration of references has been reviewed and modified accordingly

--two references "1" Pagano 2017 and Ljungman 2011?

numeration of references has been reviewed and modified accordingly

--doi is shown for only a few (2, 7, 44, 46, 48-49,52,53,78,80-81 and others) 

DOI have been included whenever available

--some are underlined (47,49,88, 90, 91-93, 99, 100, 105, 107)

references has been reviewed and modified accordingly

--journal in Italic (5) 

references has been reviewed and modified accordingly

-- reference 6 : incomplete ++

Reference 6 has been completed

--reference 7 = 85 , twice the same

References has been reviewed and modified accordingly

--some references with 3 authors et al, others with more than 6 authors listed...

references have been reviewed and modified including only the first three Authors

Round 2

Reviewer 1 Report

This manuscript can be considered for publication in the Journal of Fungi. Congratulations!